# Health Literacy among Non-Familial Caregivers of Older Adults: A Study Conducted in Tuscany (Italy)

**DOI:** 10.3390/ijerph16193771

**Published:** 2019-10-08

**Authors:** Guglielmo Bonaccorsi, Francesca Pieralli, Maddalena Innocenti, Chiara Milani, Marco Del Riccio, Martina Donzellini, Lorenzo Baggiani, Chiara Lorini

**Affiliations:** 1Department of Health Sciences, University of Florence, Viale GB Morgagni 48, 50134 Florence, Italy; guglielmo.bonaccorsi@unifi.it (G.B.); f.pieralli@gmail.com (F.P.); martinadonzellini@gmail.com (M.D.); 2School of Specialization in Hygiene and Preventive Medicine, University of Florence, Viale GB Morgagni 48, 50134 Florence, Italy; maddalena.innocenti@unifi.it (M.I.); chiara.milani@unifi.it (C.M.); marco.delriccio@unifi.it (M.D.R.); 3AUSL Toscana Centro, Florence, Piazza Santa Maria Nuova 1, 50122 Florence, Italy; lorenzo.baggiani@uslcentro.toscana.it

**Keywords:** health literacy, non-familial caregiver, older adults, care dependency, Newest Vital Sign

## Abstract

Many older adults who live at home depend on a caregiver. When familial support cannot provide the necessary care, paid caregivers are frequently hired. Health literacy (HL) is the knowledge and competence required of people to meet the complex demands of health in modern society. The aim of this study is to assess the HL level of paid non-familial caregivers who were enrolled through two different sources: from the homes of assisted people in two Tuscan health districts (first sample) and during job interviews in a home care agency operating in Florence (second sample). The two different recruitment contexts allow us to provide a broader view of the phenomenon, presenting a picture of the HL level of those who are already working and those who are looking for a new job in this field. One-on-one face-to-face interviews, which include the administration of the Newest Vital Sign (NVS) to measure HL, were conducted. Recruitment resulted in 84 caregivers in the first sample and 68 in the second sample. In the first sample, the mean age was 51.2 ± 9 years; 94% of the participants were women. A high likelihood or likelihood of inadequate HL (i.e., a low level of HL) was found in 73.8% of cases. In the second sample, the mean age was 43.7 ± 11.5 years; 83.8% of the participants were women, and 80.9% had a low level of HL. In both samples, HL was statistically associated with the level of understanding of the Italian language. In conclusion, inadequate HL is an under-recognized problem among non-familial caregivers. Educational programs that aim to increase HL skills could be an effective approach to improving the qualification of informal healthcare professionals.

## 1. Introduction

### 1.1. Care Dependency of Older Adults with Disabilities

With the increase in life expectancy and the prevalence of disabilities and comorbidity related to aging, care dependency has become more increasingly relevant for older adults, who often have a level of functional ability that is insufficient for undertaking the basic tasks necessary for daily life without the assistance of others [1]. In fact, it is estimated that the vast majority of older people will need to be cared for by at least one other person in the final years of their lives [2].

In Italy, as in other Western countries, many older people with severe disability receive care in their home, but the assistance provided by public healthcare services is minimal [3,4]. According to a recent study, Germany and Italy are the two European countries with the highest old-age (older adults) dependency ratio and the lowest percentage of people receiving long-term home care, as well as the lowest governmental expenditure in this regard [5]. 

Many older adults who live at home depend partially or totally on a caregiver. When familial support cannot provide the necessary care, many are compelled to hire a paid caregiver, i.e., a non-familial individual who receives payment for directly assisting people in their daily self-care activities within their home, including supporting the proper use of, and adherence to, complex medication regimens, setting medical appointments and accompanying older adults when they attend them, and assisting with nutrition [6]. Moreover, paid caregivers communicate directly with the seniors and their families about the health needs that they encounter [7]. Nonetheless, it is worth mentioning that, in Italy, any form of legal regulation/accreditation for work of this nature is absent.

According to recent estimates, Italy has about one million paid caregivers of older adults with disabilities, and most of them have a low educational level and low specific qualification [8,9]. Moreover, about 80% of them come from abroad, mostly from Eastern European countries (40%) and Asia (20%), and 58% have a low level of comprehension of the Italian language [8,9]. These data, as well as the lack of national regulations and standardization of duties for paid non-familial caregivers, are also observed in other countries [10,11]

In Italy, as elsewhere, private agencies and cooperatives play an important role in recruiting paid caregivers to support families with non-self-sufficient family members, although other ways of selecting caregivers exist, however diffused they may be.

### 1.2. Health Literacy: A Skill for Caregiving

Health literacy (HL) is the knowledge and competence required of people to meet the complex demands of health in modern society. According to Sørensen [12] “Health literacy is linked to literacy and entails people’s knowledge, motivation, and competences to access, understand, appraise, and apply health information in order to make judgments and take decisions in everyday life concerning healthcare, disease prevention, and health promotion to maintain or improve quality of life during the life course.” Improvement in HL can contribute significantly to the development of a new type of relationship between individuals and the health system: consideration should be given to not only the fundamental basis for therapy adherence and the patient-physician relationship but also the change that is necessary to realign the evolution of the health system and people’s knowledge and empowerment [13].

HL refers to how people comprehend health care information; it deals with an individual’s ability to read, understand, and use health care information to make effective health care decisions and follow instructions for treatment [14]. The HL level appears to undergo an aging-related decrease: the rate of decline increases with age, with people aged more than 80 years being the most vulnerable to rapid decline. Cognitive function decline due to aging appears to affect the likelihood of HL decline [15], but HL abilities, skills and experiences of other people can compensate for an individual’s poor HL: HL is not only an individual ability but also a distributed resources available within an individual’s social network. Thus, HL is considered a multidimensional construct that encompasses individual capacities, interpersonal elements, as well as the broader health system and community factors [16,17,18]. 

HL is required in order to take care of the health of others. Emerging studies suggest positive associations between low caregiver HL and poorer care recipient health outcomes; for example, poorer self-management behaviours were associated with an increase in care recipients’ use of health services [10]. If the caregiver does not have adequate HL, then compliance with the medical plan of care for the senior may be compromised, and the senior may be inadvertently harmed [11]. In a study conducted by Erickson, the HL level of the caregiver was significantly associated with medication administration tasks, suggesting that ensuring that caregivers understand medication regimens or that improving their medication-related HL (or both) may be an important step to ensuring the safe and effective use of medications by people with disabilities [19]. In a study conducted by Lindquist et al., the HL levels of caregivers were not related to the performed health-related tasks, suggesting the absence of the assessment of health-related competences prior to assigning roles involving health care [11]. This aspect is particularly relevant because community-dwelling older people with disabilities may present a high level of cognitive impairment, physical impairment, or both, which could result in a higher level of skills required to take care of them [20,21,22]. According to Garcia et al. [23], senior health-related conditions and caregiver characteristics are inter-related and have a direct effect on the patient’s health outcomes.

Moreover, some studies have also suggested that low caregiver HL negatively affects caregiver health outcomes, such as increased caregiver burden: caregivers’ HL is inversely related to their stress level and their unmet information needs [10]. 

For all these reasons, the inclusion of caregiver HL as a priority in public health policy has the potential to improve the health outcomes of both the care recipient and the caregiver. Given these premises, the knowledge of the HL levels of caregivers has become fundamental for preserving seniors’ health [24]. Nevertheless, only a few studies have investigated the HL level of this particular group, and they have found a general prevalence of low HL [10,19,23,25]. Moreover, in a study conducted by Lindquist [6], it was determined that none of the 462 investigated home care agencies performed HL assessments of potential caregivers.

### 1.3. Aim of the Study

The aim of this study is to assess the HL level of a sample of paid non-familial caregivers who were enrolled in two different settings in Tuscany (Italy): the health district of Valdinievole (Pistoia) and Florence and a home care agency operating in the province of Florence. The two different recruitment contexts allow us to provide a broader view of the phenomenon and present a picture of the HL level of those who are already working and those who are looking for a new job in this field. Specifically, the first sample allows us to measure the HL level of paid non-familial caregivers who have been hired by families with older adults with disabilities who are included in a regional social support program financed by the Tuscan regional government. For this sample, data on seniors’ needs are available. The second sample allows us to describe the HL level of caregivers who were not working with seniors at the moment of the enrolment but referred to a private agency to find employment. Since agencies are not the only means by which families find and hire a caregiver, there should be differences in the characteristics of non-familial paid caregivers by the mode of contact and hiring: thus, the data for the two samples are described separately.

## 2. Materials and Methods 

### 2.1. Study Design and Sampling Procedure

The study complies with the principles laid down in the Declaration of Helsinki and has obtained the Area Vasta Centro Ethics Committee approval (Cod. 10809_oss).

In this research, two samples of caregivers were included: a sample of non-familial caregivers who were interviewed at home, and another sample of family assistants whose HL was measured during a job interview carried out by the staff of a private job agency operating in Florence. 

The first sample was selected from non-familial caregivers working in the health districts of Valdinievole (Pistoia) and Florence and providing home assistance to non-self-sufficient citizens above 65 years (with different degrees of loss of independence). The caregivers were enrolled by contacting the families that were included in a regional social support program financed by the Tuscan regional government, which provides social and health assistance to impaired older adults and their families. This program aims to build a specific and customized healthcare project for dependent senior citizens after a multidimensional assessment of their needs is performed by a multidisciplinary team of social and healthcare professionals. A great part of the healthcare services is devoted to indirect assistance by providing a monthly sum of money to hire a caregiver. To better comprehend the context in which the survey was conducted, it is important to note that in Italy, as in other European countries, although the policies and the financing of long-term home care are established at a national level, each region is responsible for delivering healthcare services on its own [26]. The social and healthcare managers of the health districts of Valdinievole and Florence provided a list of older adults receiving economic help to hire a caregiver. After being informed about the aim of the study and agreeing to take part in it, each older adult or one of his/her family members provided the non-familial caregiver’s contact information; subsequently, each of the paid caregivers was contacted for an interview. Written consent was obtained from both the non-familial caregivers and the senior/family member. During the study period, 303 older adults were receiving economic assistance. Among the paid caregivers hired by the families of those older adults, 84 agreed to be enrolled and signed the informed consent (compliance: 28%). Of the others, 39% refused to participate, 23% were untraceable, and 17% were no longer working for the contacted family because of the death or institutionalization of the senior.

The second sample was composed of non-familial caregivers recruited during job interviews by a private job agency operating in the field of domestic older adult healthcare, with a catchment area corresponding to the Florence health district. Specifically, during the study period, each caregiver who underwent a job interview was informed about the aim of the study and was asked to take part in it. The number of enrolled caregivers in this sample was 68, with compliance equal to 97% (two caregivers refused to participate).

### 2.2. Health Literacy Measurement

For both samples, HL levels were measured by means of the validated Italian version of the Newest Vital Sign (NVS) [27,28,29]. This is a tool based on the information presented on a nutritional label, printed on a sheet, that the caregiver consults during the test administration. The NVS measures literacy (the ability to perform basic reading tasks), comprehension (the ability to derive meaning from sources of information), numeracy (the ability to perform basic numerical tasks and arithmetic operations), application/function (the ability to use, process or act on health-related information and apply new information to changing circumstances), and evaluation skills (the ability to filter, interpret and evaluate information) [30]. It consists of seven questions regarding nutrition facts and is administered during a direct interview. For example, the first question is, “How many calories (kcal) will you eat if you eat the whole container?”; the fifth question is, “Imagine that you are allergic to the following substances: penicillin, peanuts, latex gloves, and bee stings. Is it safe for you to eat this ice cream?”. With an objective assessment expressed as a final score ranging from 0 to 6, this quick tool (administered in approximately five minutes) classifies the participants into one of three categories: high likelihood of limited HL (score: 0–1); possibility of limited HL (score: 2–3); and adequate HL (score: 4–6). People classified in either the intermediate category (possibility of limited HL) or the lowest one (high likelihood of limited HL) are at risk of insufficient HL. The NVS was first developed to measure HL in a clinical setting and it has also been applied in other contexts, such as in population-based studies [28,29,30,31,32,33].

### 2.3. Data Collection in the Health Districts of Florence and Valdinievole

Data related to paid non-familial caregivers were collected through face-to-face interviews using an ad hoc questionnaire. The questionnaire included questions related to socio-demographic characteristics (age, gender, country of origin, years in Italy, education level), how the caregiver found the job (e.g., through an agency, church, family contacts, word of mouth), the number of working hours, and the duties performed (e.g., older adult care, drug administration, support in the use of medical devices, household help), in addition to the NVS. 

For caregivers who were not native Italians, the interviewer—at the end of the interview—made a subjective judgement of each caregiver’s level of understanding of the Italian language (low, medium, or high).

The HL levels of the caregivers were compared with the characteristics of the assisted older adults, by obtaining data about the social and health conditions of seniors from the multidimensional assessment of the health district. The assessment was led by a multi-professional team of social and healthcare workers, including a nurse, a physician and a social worker, and ultimately supported by geriatricians and physiatrists. 

The team collected information for each assessed subject by using internationally validated scales: the dependency level in basic activities of daily living (BADLs) by means of the Minimum Data Set Activities of Daily Living (MDS-ADL) Long-form Scale [34]; the dependency level in instrumental activities of daily living (IADLs) using Lawton’s IADL Scale [35]; the presence of impairment in cognitive function using the short portable mental status questionnaire (Pfeiffer’s test) [36]; and the presence of mood disorders by means of the Minimum Data Set–Home Care (MDS-HC) [37]. Moreover, using the same scales, the comprehensive level of social and healthcare needs was estimated using a multidimensional assessment of the degree of dependence, on a scale of 1–5, where 5 indicates the highest level of care needed [22,38]. 

### 2.4. Data Collected by the Job Agency 

Data were sent anonymously from the agency by the staff in charge with whom the research group was in contact. During the job interview, the following variables were collected: age, sex, country of origin, educational level, tasks carried out provided in previous work experiences, typology of participants assisted in previous work experiences, work experience (years), linguistic (Italian) skills, ability to understand Italian and HL (NVS test score).

In particular, with regard to previous experience, the ability to ensure the person’s hygiene (in bed and in the bathroom), the ability to provide support for walking, the ability to administer drugs, and first-aid knowledge were investigated. The typology of patients assisted in previous experiences concerned different categories of older adults (e.g., not self-sufficient, suffering from Alzheimer’s or Parkinson’s disease, and suffering from depression or senile dementia).

### 2.5. Statistical Analyses

The data of the two samples of family assistants (interviewed at the elderly people’s homes or at the job agency) were collected in an electronic database and analysed using IBM SPSS 25^TM^, (Armonk, NY, US). The information was subsequently anonymised by assigning a single and unique numeric code. In particular, for the data collection and the subsequent statistical analysis in the health districts of Florence and Valdinievole, a single and unique numeric code was generated to each dyad of non-familial caregiver-senior in order to link the data of each non-familial caregiver to those of the senior whom he/she assists. Any original, identifiable information was destroyed when the study was completed.

Data are presented as the mean ± standard deviation or as a percentage. For each variable, normality was assessed using the Kolmogorov-Smirnov test.

Associations between the caregivers’ HL levels were classified in two levels (“high likelihood of limited HL” and “possibility of limited HL” compared with adequate HL). The other variables related to the non-familial caregivers, and the characteristics of the assisted older adults were assessed using Fisher’s exact test for categorical data. Student’s two-tailed t-test for independent data and the Mann–Whitney U test for continuous data were respectively used for normally distributed and non-normally distributed continuous data. Spearman’s rank correlation coefficient was used as a nonparametric measure of rank correlation between HL (NVS score) and data related to the caregivers and older adults, which were reported as continuous variables. For each analysis, an alpha level of 0.05 was considered significant.

## 3. Results

### 3.1. Caregivers Interviewed at Home (First Sample)

The number of caregivers interviewed at home is 84, with a mean age of 51.2 ± 9.7 years. Most of them (88%) had been hired directly by the family from word-of-mouth while 5% had been hired by a job agency. Table 1 reports the characteristics of non-familial caregivers and seniors, stratified by the HL categories of the caregivers. The vast majority is formed by women (94.0%). Only two are Italian; most of the others come from Romania (48.8%) and other Eastern European countries. Of the non-familial caregivers coming from abroad, 85.4% have a good (medium or high) comprehension of the Italian language, which aligns with their mean length of stay in Italy (10.7 ± 5.2 years). The non-familial caregivers reported their level of education as 11.3 ± 3.7 years of schooling; 23.8% have a degree, 42.9% are high school graduates, while 29.8% attended only primary or secondary school. Most non-familial caregivers (88.1%) gained employment with the seniors directly through family contacts, and the type of job is full-time in 63.1% of the cases. For all of them, job duties include drug administration, personal care, household management, and food preparation and administration. According to the NVS score, 26.2% of the caregivers have adequate HL (NVS score: 4–6), and 73.8% have some limitations in HL (34.5% have the possibility of limited HL, and 39.3% have a high likelihood of limited HL).

The mean age of the participants is 87.7 ± 7.7 years and 82.1% of them are women. The dependency level in BADLs was measured by the MDS-ADL Long-form Scale, which reports a score of 18.9 ± 5.9. According to Pfeiffer’s test, 64.3% of the older adults have moderate/severe cognitive impairment. The level of personal care needs was defined by the regional scale validated in the Tuscan context, and the results indicate that 38.1% are classified at Level 4, that is, a severe degree of impairment. 

The HL levels of the non-familial caregivers are statistically associated with their level of understanding of the Italian language (for foreigners, *p* < 0.001): the higher the level of comprehension, the higher the percentage of caregivers with adequate HL or the higher the NVS score (Table 2).

The results show no statistically significant association between the caregivers’ HL and the seniors’ intellectual or physical impairment. The correlation analysis reveals a significant correlation between HL (NVS score) and the years of schooling (r = 0.281): HL increases with an increase in the years of schooling.

### 3.2. Caregivers Recruited During the Job Interview (Second Sample)

The sample consists of 68 participants (Table 3). The mean age is 43.7 ± 11.5 years. Only six non-familial caregivers (8.8%) were born in Italy: the majority were born abroad. The most represented geographical areas are Central and Southern America, which are the origins of 26 non-familial caregivers; more specifically, 20 caregivers are from Peru (29.4% of the sample). The greater part of the sample is formed by women (83.8%). Regarding the type of services performed and the type of participants assisted in their previous work experiences, the results show that the majority of the interviewed caregivers (77.9%) have had previous experience with non-self-sufficient older adults. In most of the cases, the care provided includes complex duties, such as drug therapy supervision and first aid (86.8%). Assistance in walking and in personal hygiene was reported in 82.4% of cases. Regarding their working experience in this field, almost 27.9% of the caregivers reported having 1–3 years of practice; 29.4% reported 2–7 years, and 17.6% reported having experience of more than 7 years. The information on years of schooling was not processed because of missing answers for 77.9% of the interviewed participants. The opinion on the participant’s ability to speak and understand the Italian language (for foreigners), was given by the interviewer as one of three levels (low, medium or high), and the results show an exact correspondence between the level of expression and the level of comprehension for all the subjects. In particular, only six non-familial caregivers from abroad have only a low level of expression and understanding; for the majority (86.8%), the level was judged by the recruiting agency as good. Further, the results show that 80.9% of the sample has a low level of HL. 

The HL of the non-familial caregivers is higher when the level of understanding of the Italian language is higher, although the relationship is at the limit of statistical significance (*p* = 0.07) for HL levels (Table 4). No significant association emerges between the HL level and previous working experience in terms of duties provided and the characteristics—i.e., the degree of loss of independence and the need of the older adults for assistance.

## 4. Discussion

Given the current situation, which is characterized by a diffused economic crisis in the public welfare systems in many countries and, consequently, a decrease in the public healthcare services provided by the country or the region in question, older adults in Italy—even with various degrees of self-insufficiency—are mainly being assisted at home, with an increasing usage of non-familial individuals being hired as caregivers. From this perspective, the role of non-familial caregivers in the care of older adults is becoming increasingly important because their quality of life greatly depends on the skills and duties of the caregiver [2]. In Tuscany, as in most other areas in which a demographic and epidemiological transition is in progress, this situation has been handled by the regional government, which, by means of specific legislation, is trying to provide effective solutions to ensure good healthcare for elderly people while keeping them at home instead of institutionalizing them [39,40]. In this framework, the role of paid non-familial caregivers acquires greater importance for ensuring the continuity of care. 

It should be noted that elderly people usually need healthcare support (drug administration, use of medical devices); thus, paid caregivers effectively act as informal healthcare workers. Since they perform tasks that have a direct effect on the patient’s health status [23], caregivers’ low HL level may have a negative impact on health outcomes; compliance with the medical plan of care may be compromised, and the seniors may inadvertently be harmed [11]. 

Consistent with previous studies [11,23], this research reveals that many caregivers—who were assessed either at seniors’ homes or during the job interview—show a high likelihood or possibility of limited HL.

In our study, HL (NVS score) is significantly correlated with the non-familial caregivers’ years of schooling. This is in accordance with the recent definition of HL, which includes skills regarding the capability of accessing, understanding, appraising, and applying health information [12].

Other authors have already described the association between the level of acculturation and the level of HL in caregivers for seniors [23]. An individual’s HL may fluctuate with different contexts and is also dynamic, changing over time and with varying circumstances. In this regard, an individual may be considered health literate in one culture or context but not in another [41]. The level of understanding of the Italian language, which is a demographic determinant of HL belonging to the area of acculturation and assimilation, proves to be an antecedent that is significantly associated with the HL level [23], even if the NVS measures only the functional dimension of HL. Since paid caregivers communicate directly with older adults or their families about the health needs that they encounter rather than discussing their concerns with the health care team [7], a language barrier is actually a critical obstacle to providing assistance and can compromise the quality of care. This is the reason that, in this study, the NVS was administered in Italian and not in the mother language of the caregiver: in fact, the aim was to assess the caregiver’s real capacity to interact with the senior and his or her family members regarding healthcare questions. On the other hand, a language barrier itself could have affected the results of the NVS test and not necessarily reflect the caregiver’s real healthcare competencies. In such a situation, the introduction of language courses for paid foreign caregivers could result in the improvement of healthcare competencies and higher NVS scores, with important results in terms of the effectiveness of older adult care. Moreover, it is important to note that language is the most appropriate and effective way to approach people with dementia, so communication skills are fundamental to caregiving [7,42]. 

As in previous studies [2], the results of our study show that the majority of paid caregivers for seniors are foreign workers without high levels of education. This could be attributed to the absence of any form of legal regulation/accreditation to manage the hiring of such workers. This absence of norms implies that, at the beginning of the employment relationship, very little is known about the caregivers’ skills. The choice is dependent more on economic advantages—the cheaper the caregiver, the greater the possibility of hiring him/her—than on an assessment of the competencies needed to effectively assist the senior. 

As already experienced positively in other contexts [43], it might be necessary for at least a great majority of non-familial caregivers to receive adequate training before proceeding to take care of older adults. This should result in the protection of families from potential damages owing to caregivers’ incapacity to fulfil their role, as well as in the provision of an opportunity for caregivers to enhance their abilities and skills. On the part of the health system and the families of the seniors, testing a potential caregiver’s HL could be a way to identify training needs prior to assigning him/her a healthcare role. This would facilitate tailoring interventions and responsibilities to the needs of the assisted people, thus providing them with optimal care. However, many recent studies have shown that, currently, caregivers may not be adequately prepared for all that the position requires of them when caring for an older adult [11,44,45].

It may be advisable to raise the awareness of family members and the whole domestic healthcare system about the need to improve the quality of caregivers’ assistance by training the people hired for this job.

The study of the senior–caregiver dyad—in which the healthcare needs of the older adults are compared with the HL of the caregiver—can effectively serve as a crucial assessment of the service provider. This would help family members decide on a caregiver in light of the required assistance skills to prevent negative outcomes for the senior.

The results reveal that 38.1% of the study participants have a high level (the fourth degree, according to the regional classification assignment) of personal care needs, confirming the earlier conclusion: a high dependence in BADLs, cognitive impairment, and mood or behavioural disorders require a high level of assistance [22] from the ‘informal’ sector of healthcare, that is, the professional area in which these caregivers work. 

Although the results of our study are consistent with those of the HL literature, some limitations exist. First, it is a convenience sample that comprises caregivers who assist patients who are included in a specific regional program of social support. Second, the sample size is relatively small owing to difficulties obtaining two different forms of contact and consent (from the older adult or one of his/her relatives as well as from the caregiver), resulting in very difficult recruitment. Third, the use of the NVS alone, without pairing results to a test that is able to measure the interactive and critical dimensions of HL, limits the assessment of the HL skills of these workers. 

Despite these limitations, there are some statistically significant results, suggesting that the findings are relevant and deserve further investigation. For this reason, we are planning to extend recruitment to other areas near Florence.

## 5. Conclusions

Inadequate HL is an under-recognized problem among paid non-familial caregivers of older adults. The results of this study indicate that many caregivers have a high likelihood or possibility of limited HL, although they are expected to provide health-related services for the older adults whom they assist. The prevalence of low HL among caregivers highlights the need for clinicians and other healthcare professionals who work with older adults in a home care setting to apply universal HL precautions, not only for patients but also for their caregivers.

## Figures and Tables

**Table 1 ijerph-16-03771-t001:** The sample interviewed at home: descriptive analysis (N = 84).

**Variables**		**N**	**%**
Gender	Females	79	94.0
	Males	5	6.0
Country of origin	Italy	2	2.4
	African countries	4	4.8
	American countries	9	10.7
	Asian countries	5	6
	European countries (other than Italy)	64	76.2
Educational level	None	1	1.2
	Primary school	5	6.0
	Middle school	20	23.8
	High school	36	42.9
	Graduate	20	23.8
Domicile with the senior	Yes	73	86.9
Level of comprehension of Italian language (for foreigners, N = 82)	Low	12	14.6
	Medium	30	36.6
	High	40	48.8
Health Literacy	High likelihood of limited HL	33	39.3
	Possibility of limited HL	29	34.5
	Adequate HL	22	26.2
**Variables**	**Mean ± SD**	**Median**	**Range**
Age (years)	51.2 ± 9.7	54.0	25–66
Years in Italy (for foreigners)	10.7 ± 5.2	10.0	2–27
Years of schooling	11.3 ± 3.7	12	0–18
Health literacy (NVS score)	2.6 ± 2.1	3.0	0–6

**Table 2 ijerph-16-03771-t002:** The sample interviewed at home: health literacy category by the level of comprehension of the Italian language (for foreigners, N = 82). For the health literacy category, χ^2^ test: *p* < 0.001. For the NVS score, Kruskal–Wallis test: *p* < 0.001.

Level of Comprehension of Italian Language	Health Literacy Category N (%)			Total N (%)	Health literacy (NVS score) Mean (SD); median
	High likelihood of limited HL	Possibility of limited HL	High likelihood of adequate HL		
**Low**	11 (91.7)	1 (8.3)	0 (0)	12 (100)	0.4 (0.8); 0
**Medium**	13 (43.3)	12 (40.0)	5 (16.7)	30 (100)	2.2 (1.9); 2.5
**High**	8 (20.0)	15 (37.5)	17 (42.5)	40 (100)	3.6 (1.9); 4
**Total**	32 (39.0)	28 (34.1)	22 (26.8)	82 (100)	2.6 (2.1); 3.0

**Table 3 ijerph-16-03771-t003:** The sample recruited during the job interview: descriptive analysis (N = 68).

**Variables**		**N**	**%**
Gender	Females	57	83.8
	Males	11	16.2
Country of origin *	Italy	6	8.8
	African countries	6	8.8
	American countries	26	38.2
	Asian countries	4	5.9
	European countries (other than Italy)	25	36.8
Years of experience *	<1	13	19.1
	1–3	19	27.9
	3–7	20	29.4
	>7	12	17.6
Caregiving skills	Personal hygiene	56	82.4
	Walking assistance	56	82.4
	First aid/help with medication management	59	86.8
Kind of seniors assisted in previous jobs	Not self-sufficient	53	77.9
	Alzheimer’s disease	33	48.5
	Immobilization	38	55.9
	Dementia	22	32.4
	Depression	15	22,1
	Parkinson’s disease	14	20.6
Level of comprehension of Italian language (for foreigners, N = 61) *	Low	6	9.8
	Medium	24	39.3
	High	29	47.5
Health literacy category	High likelihood of limited HL	25	36.8
	Possibility of limited HL	30	44.1
	Adequate HL	13	19.1
**Variables**	**Mean ± SD**	**Median**	**Range**
Age (years)	43.7 ± 9.7	43.0	21–67
Health literacy (NVS score)	2.3 ± 1.5	2.0	1–6

* Missing data: 1 for “Country of Origin”; 4 for “Years of experience”; 2 for “Level of comprehension of the Italian language”.

**Table 4 ijerph-16-03771-t004:** The sample recruited during the job interview: health literacy by the level of comprehension of the Italian language (for foreigners, N = 59). For the health literacy category, χ^2^ test: *p* = 0.07. For the NVS score, Kruskal–Wallis test: *p* = 0.005.

Level of Comprehension of Italian Language	Health Literacy Category N (%)			Total N (%)	Health Literacy (NVS Score) Mean (SD); Median
	High Likelihood of Limited HL	Possibility of limited HL	High Likelihood of Adequate HL		
**Low**	5 (83.3)	1 (16.7)	0 (0)	6 (100)	1.3 (0.8); 1
**Medium**	12 (50.0)	10 (41.7)	2 (8.3)	24 (100)	1.8 (1.5); 1
**High**	8 (27.6)	14 (48.3)	7 (24.1)	29 (100)	2.8 (1.6); 2
**Total**	25 (42.4)	25 (42.4)	9 (42.4)	59 (100)	2.3 (1.5); 2.0

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
