# Peer review of "Health Literacy among Non-Familial Caregivers of Older Adults: A Study Conducted in Tuscany (Italy)"

_ijerph, 2019, doi:10.3390/ijerph16193771_

Round 1
Reviewer 1 Report
Thank you for the opportunity to review this clearly written, well-expressed paper. I have numbered my comments for the authors below.
Line 37: 'occidental' not commonly used word, sounds a bit archaic. Would replace with other descriptor (western, industrialized, high-income) or remove phrase, i.e. 'In Italy many severely disabled....'
Line 47: Sentence trails off - word(s) missing
Throughout the paper I was wondering if any formal qualifications required to work as a home carer in Italy - this info is given at line 278, but I think could be useful to include this as background in the Introduction
I'm not sure that I understand the utility/relevance in collecting and reporting data about the characteristics of the seniors who were being cared for, particularly as they were only collected for part of the sample. These are only briefly mentioned in the results to say that there was no association between HL and seniors' level of impairment. Did they authors expect to find an association? If so, why? I think that including this data requires stronger justification.
Following from the previous comment, stating at Line 300 that 38% of the senior population has a high level of care needs, based on the study sample is a strong statement to make given that 70+% of the possible sample did not participate. It isn't completely clear that you're only talking about this small sample, and not the senior population as whole.
It's hardly surprising that HL was associated with language proficiency, given that the NVS is essentially a comprehension test, which measures only one dimension of HL (functional) in one context, and does not include other essential dimensions, including interactive/communicative and critical HL, or acknowledge the context-specific nature of HL. You should elaborate on the limitations of the NVS to measure HL and its conceptual dimensions
Author Response
We want to thank the reviewer for the comments and suggestions. Here the point-by-point reply.
Line 37: 'occidental' not commonly used word, sounds a bit archaic. Would replace with other descriptor (western, industrialized, high-income) or remove phrase, i.e. 'In Italy many severely disabled....'
We have modified the word as requested.
Line 47: Sentence trails off - word(s) missing
We have corrected the sentence
Throughout the paper I was wondering if any formal qualifications required to work as a home carer in Italy - this info is given at line 278, but I think could be useful to include this as background in the Introduction
We have added this information also in the Introduction.
I'm not sure that I understand the utility/relevance in collecting and reporting data about the characteristics of the seniors who were being cared for, particularly as they were only collected for part of the sample. These are only briefly mentioned in the results to say that there was no association between HL and seniors' level of impairment. Did they authors expect to find an association? If so, why? I think that including this data requires stronger justification.
In the Methods, we have added a sentence to better explain this concept and recalled a citation in the references.
Now, the paragraph is as following: “According to Garcia et al. [17], senior health-related conditions and caregiver characteristics have relationships with each other as well as a direct effect on the patient’s health outcomes. For this reason, to compare the HL levels of the caregivers to the characteristics of the assisted elderly people, data about social and health conditions of seniors were obtained from the multidimensional assessment of the health district, led by a multi-professional team of social and healthcare workers, including a nurse, a physician, and a social worker, eventually supported by geriatricians and physiatrists.”
Following from the previous comment, stating at Line 300 that 38% of the senior population has a high level of care needs, based on the study sample is a strong statement to make given that 70+% of the possible sample did not participate. It isn't completely clear that you're only talking about this small sample, and not the senior population as whole.
We have better specified that this percentage is referred to the sample and not to the senior general population
It's hardly surprising that HL was associated with language proficiency, given that the NVS is essentially a comprehension test, which measures only one dimension of HL (functional) in one context, and does not include other essential dimensions, including interactive/communicative and critical HL, or acknowledge the context-specific nature of HL. You should elaborate on the limitations of the NVS to measure HL and its conceptual dimensions
Thank you very much for this comment. In the Discussion, we have specified that NVS measures only the functional dimension of HL, and that this is a limitation of the study, adding that a second test would have been useful to measure also interactive and critical dimensions.
Reviewer 2 Report
The authors address a common concern as all populations report approximately 1/3 of adults have low health literacy with another 1/3 marginally affected. This manuscript is unique in that it addresses the caregivers of patients in need and some of the caregivers do not share a native language.
My first recommendation that will change your study throughout is to eliminate the Study 1 you address as you do not have evaluation data for that cohort; focus on study two- the size of the sample does not bother me as this is a topic that has not been studied at great detail.
Additionally, it appears when reading the manuscript that authors divided up sections and gave no thought the flow of the entire document. I encourage you to have a professional editor review the paper for grammar, flow and conciseness after you eliminate the information about Study 1.
Author Response
We want to thank the reviewer for the comments and suggestions. Here the point-by-point reply.
The authors address a common concern as all populations report approximately 1/3 of adults have low health literacy with another 1/3 marginally affected. This manuscript is unique in that it addresses the caregivers of patients in need and some of the caregivers do not share a native language.
My first recommendation that will change your study throughout is to eliminate the Study 1 you address as you do not have evaluation data for that cohort; focus on study two- the size of the sample does not bother me as this is a topic that has not been studied at great detail.
Thank you very much for this comment. We have discussed a lot about your suggestion, but finally we have decided - also on the basis of the first reviewer’s comment – not to eliminate the data referred to the first sample (that is, the paid caregivers recruited at home). In fact, it gave us information about the relationship of the dyad “caregiver vs senior”, while the second sample was formed by persons recruited during the job interview. In line with Garcia et al, senior health-related conditions and caregiver characteristics have relationships with each other as well as a direct effect on the patient’s health outcomes.
Additionally, it appears when reading the manuscript that authors divided up sections and gave no thought the flow of the entire document. I encourage you to have a professional editor review the paper for grammar, flow and conciseness after you eliminate the information about Study 1.
We have sent the article to a professional editor to revise grammar, flow and conciseness. Nonetheless, the division in sections is required by the Journal, as expressed in the Instructions for Authors.